# Applying Scrum in an Online Physics II Undergraduate Course: Effect on Student Progression and Soft Skills Development

Emmanuel Lourakis * and Konstantinos Petridis

Department of Electronic Engineering, School of Engineering, Hellenic Mediterranean University, 73100 Chania, Greece
* Correspondence: lourakismanolis@gmail.com

**Abstract:** The everchanging higher education landscape dictates the need for innovative, engaging, and efficient teaching methods along with promoting soft skills among the students. Scrum's framework for effective group work and its methodology tools is a perfect fit to this end since it has been previously applied in higher education for enhancing collaborative learning and developing skills such as communication, teamwork, and problem solving. This paper describes how Scrum was used remotely for a small portion of the students enrolled in an online Physics II undergraduate course during the pandemic lockdown period of an Electronic Engineering School. The primary goal was to investigate the feasibility of Scrum under remote teaching conditions as a facilitator of collaborative group learning. A secondary objective was to observe the students' interactive engagement in distance learning, along with the possible advancement of soft skills. The paper concludes with the result analysis of the feedback collected from the teachers and the students involved in this study.

**Keywords:** Scrum; higher education; distance learning; active learning; soft skills; group work

## 1. Introduction

Agile has become, in recent years, a predominant development concept and the preferred approach for many development and production teams, especially those trying to create a culture of continuous delivery. Before the emergence of Agile, production teams (especially those in the software, aerospace, manufacturing, and defense industries) would plan the entire project following the Waterfall concept. This approach follows a pre-determined path in which teams initially set project requirements and the scope of work and then design a product on the basis of those preset requirements. The finished product is launched after fixing any problems discovered during testing. Even though this approach seems fine, the developers' teams needed to adhere to the requirements and scope of work initially set out at the very beginning of the project and could not alter or amend anything along the way. The problem with this approach arises from the fact that it could take years before teams finished the task at hand. During those years, the nature of the situation would often change (but the project requirements would not), rendering the planned solution out of date by the time it finally got to market. With this problem in mind, several development teams during the 1990s began to change their approach to planning and delivering new products, using methods such as Scrum, Rapid Application Development, Extreme Programming, Feature-Driven Development, and Pragmatic Programming [1]. These approaches to software development are the earliest methods in the history of Agile that ultimately led to what we know today as the Agile Manifesto [2]. According to the values of Agile, individuals and interactions take priority over processes and tools, and working software has priority over comprehensive documentation. At the same time, customer collaboration is more important than contract negotiation, and responding to change is prioritized over following a plan.

Scrum is one of several Agile frameworks depicted in the Agile Subway Map by the Agile Alliance, a global non-profit membership organization founded on the Manifesto for Agile Software Development [3]. It is a lightweight framework that helps people, teams, and organizations generate value through adaptive solutions for complex problems. Scrum employs an iterative, incremental approach to maximize predictability and avoid unnecessary risks to a project. It is based on empiricism and lean thinking and offers the applicants the freedom to work as they like while embracing the values and pillars of the framework [4].

Agile methodologies are proven effective in higher education as teaching/learning practices based on the best concepts and ideas from software engineering and software development [5]. Especially, the Scrum framework is proven to facilitate the level of engagement required in group projects, even when applied in higher education (HE) teaching and learning processes, mainly because it enables active learning and self-management [6–9]. In the active learning concept, the learners are responsible for their own learning [10–14], and this practice is well served by the principles of the agile manifesto since it values student-driven inquiry and continuous improvement among other values [15]. By design, the Scrum framework also enhances the collaborative learning of the group when the desired product is knowledge since its core structure aims to maximize the team's performance over time. Previous research in both face-to-face and online distance education proves the benefits of cooperative learning for undergraduate students [16,17].

The bibliographic research implemented showed that the existing publications related to the application of Scrum in Higher Education [6–8] deviated from the main guidelines of Scrum as described in the 2020 update of the Definitive Scrum Guide by Ken Schwaber and Jeff Sutherland. The diversions from the Definitive Scrum Guide [4] were as follows:

- Daily Scrum meetings did not occur every working day.
- Unclear presentation of the construction of the Product Backlog.
- Sprint Retrospect meetings were not held.
- Delivery of Increments between the Sprints and not in the Sprint Review.
- Unclear references to the Sprint Planning meeting.
- Extra evaluation and planning processes were added apart from those described in the framework.
- Duties and roles were intermingled.

The deviations as mentioned above, according to the authors, lower the impact of Scrum on the level of students' devotion to the project (Daily Scrum meetings did not occur every working day) and degrade the agile characteristics of Scrum (Sprint Retrospect meetings were not held, vague references to the Sprint Planning meeting, delivery of increments between the Sprints and not in the Sprint Review, and extra evaluation processes). The author's research question aims to examine the impact of Scrum framework on students' understanding through collaborative-based learning. The main assumptions we adopted are based on the students' first-time exposure (a) to student-centered learning approaches and (b) to real-life working conditions.

The authors support the Scrum framework because of (a) enhanced induced cooperation and collaboration among the functional teams (especially during the isolation period of the COVID-19 era); (b) its natural connection to the market world working conditions and demands; (c) its agile characteristics that enable trainers to monitor, intervene, and facilitate students' learning achievements in the right moment; and (d) the higher induced feeling of ownership of students' learning achievements. Focusing on higher education Physics I and II courses, active learning methods, such as the Active Learning Studio Model, enhanced understanding of physics concepts and developed the communication and problem-solving skills of the students [18]. To support this statement, the authors' main objective was for the students not only to learn physics concepts and laws during a distance-teaching semester but also to be able to link them with real electric and electronic devices using the Scrum framework. This is a novel approach since, to the extent of the authors' knowledge, Scrum

has never been applied before in a Physics II course as an active learning method, either in face-to-face or remote online teaching.

This work describes how this was accomplished by creating a more extensive engagement concerning their online lectures—by transferring the responsibility of their learning onto them. The path towards that goal was through cooperation, collaboration, and research with self-managed peers, as the Scrum framework advocates. In addition to the above-described purposes, a soft-skill enhancement for the students was desired. This parallel aspect of this study was inspired not only by the Scrum core pillars and values, as the Definitive Guide describes, but also from previous work in HE that has shown that Scrum can actually evolve students' abilities, affecting various areas of soft skills, such as oral presentation, punctuality, transparency, self-management, critical thinking, and searching for, processing, and analyzing information from multiple sources [19–21]. Moreover, the familiarization of the students with the actual Scrum framework was an additional positive side benefit for their future working environment since the electronic engineering sector involves project teamwork and software development.

## 2. Materials and Methods

### 2.1. Theoretical Background

Scrum framework encompasses both the objectives of active learning and engagement since it involves groups of people who collectively have all the skills and expertise to do the work and share or acquire such skills as needed [4]. Of paramount importance are the three pillars and the five values that must be embraced in parallel with the three roles, the five events, and the three artifacts.

2.1.1. Scrum Framework Pillars and Values

The first pillar is the transparency of the work that will lead to the second pillar of inspection which, in turn, will drive the third pillar of adaptation. More particularly, the terms transparency, inspection, and adaptation are defined as follows: (a) Transparency: the emergent progress and work must be visible to those performing the work (e.g., students) and to those receiving the work (e.g., the academic supervisor); (b) Inspection: the Scrum artifacts and the progress towards the agreed goals must be monitored frequently and diligently to be able to detect potential deviations and problems; and (c) Adaptation: In case of deviations beyond the accepted limits, the process being applied or the products produced must be adjusted. The necessary adjustments must occur as soon as possible to minimize the possible deviation. The entire organization must adhere to the three pillars for Scrum to be effective and successful. However, for those pillars to come to life, the Scrum team (Product Owner, Scrum Master, and Developers) must embody the five values of the framework. These values are openness, respect, courage, focus, and commitment. Openness means that everybody must be able to speak their mind on everything, from an idea for the backlog to a problem the team member faces. This cannot occur unless this person dares to step up and speak truth to power, propose a new idea, or raise an impediment. Therefore, everybody must respect everyone on the Scrum Team, so no one is afraid of peer criticism when they express their opinion. When the above values are embraced, the team can build agreement on what they should do and how they get there, enabling the team to focus on the goal. With the proper focus on the agreed goal, it is much easier for the team to commit to the project [22].

2.1.2. The Three Roles

There are three distinct roles in the Scrum framework. The Product Owner is solely responsible for the outcome and is the customer's voice that sets the vision and the priorities for the Product Goal. The Scrum Master is a leader who serves and oversees the application of Scrum at all levels of the organization. A Scrum Master's purpose is to facilitate the application of the framework and the continuous improvement of the developer's team. Developers, as the last role, are the group of people who perform the work towards the

completion of the product. The members are cross-functional and possess all the skills needed for the job. They are self-managed, with no actual leader, and decide collectively on every aspect of the work to be carried out, following the iterative manner of the five events. These three roles constitute the Scrum Team.

### 2.1.3. Scrum Events

The project starts with the event of Sprint Planning meeting, where the Scrum team under the direction of the Product Owner composes the Product Backlog, which is the list of to-do items for the whole project. After the Product Backlog construction, they collectively decide when the project is finished by stating the Definition of Done. Then, the team decides which of the product backlog items will be worked on in the upcoming Sprint, and this smaller list is the Sprint Backlog. After this meeting, the work starts for a predefined period called the Sprint, the main event and heart of Scrum. At the beginning of each workday, the third event happens under the directions of the Scrum Master, and this is the Daily Scrum. It is a meeting that should last less than 15 minutes, and the purpose is for the Scrum Master to remove any impediments the team may encounter, and after a short discussion with the developers, to re-plan or adapt the work for the rest of the day. The Sprint outcome is reviewed by the team in an event called Sprint Review, where after inspection and assessment of the completed work, it will be decided whether a product increment is produced. The fifth event is the Sprint Retrospect, which closes the iteration circle of the Scrum. In this meeting with the Developers and the Scrum Master only, the scope is to discuss the tools, processes, and interactions that took place during the Sprint and to decide which to keep and which to discard for the team to improve in the next Sprint. In each iteration, production of the three artifacts transpires. The iteration cycle of Scrum is depicted in Figure 1.

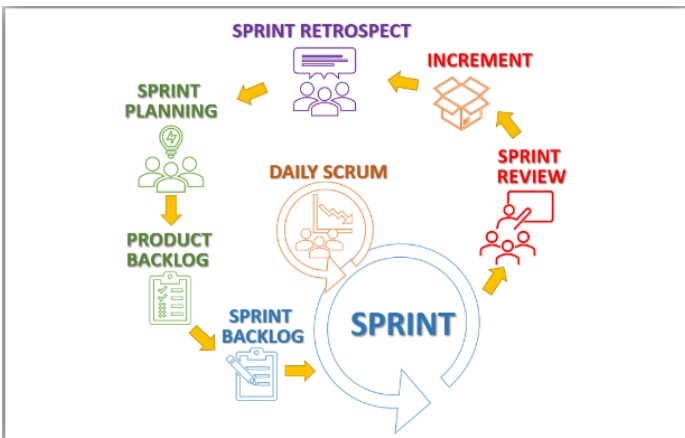

**Figure 1.** Scrum iteration cycle.

### 2.1.4. Scrum Artifacts

The first one, the Product Backlog, is the written form of the vision and the priorities leading to the final product. Although this is decided at the first Sprint Planning, it can and must be refined in each cycle. The Product Owner is responsible for prioritizing the items listed there, but every team member can add items at any time. Only the Product Owner can delete items from the Product Backlog. The second artifact is the Sprint Backlog, a subset of the Product Backlog. The last artifact is the sum of each Sprint's completed work, which gives value and leads to the Definition of Done as stated in the Sprint Planning. This last artifact is called Increment. According to Scrum, the sum of the Increments leads to the Final Product.

*2.2. Methodology*

The project was conducted on a voluntary basis in the undergraduate Physics II course of the second semester of the Electronics Engineering Department of the Hellenic Mediterranean University of Greece. The novelty was in the online implementation of the framework in a natural science course, after remote teaching conditions were imposed due to the COVID-19 pandemic lockdown. After two comprehensive online lectures regarding the Scrum guide and the framework, ten students enrolled in the project, knowing that this project was a real-time experiment in developing key soft skills, such as collaboration, decision-making, critical thinking, and time management. In order to reward the participant students and motivate the engagement of more of them in the future, an extra bonus was given. The prerequisite for receiving the bonus was the success in the semester's exams. The project was conducted in two phases throughout the semester, with two different team setups for each phase. The purpose of the two phases was to compare the effort needed from the Scrum master to facilitate a different number of teams.

2.2.1. The Roles of Product Owner and Scrum Master

The teaching professor of the class fulfilled the role of the Product Owner, and the part of the Scrum Master was assumed by a Ph.D. student of the same professor. The Product Owner inspected the job completed at each Sprint Review and provided constructive feedback to be utilized in the next Sprint by the students. On the other hand, the role of the Scrum Master was more interactive and frequent during the Daily Scrums. The Scrum Master was the 'facilitator' of the Scrum Teams towards the implementation of the artifacts by checking three elements: what has been accomplished, what are the obstacles, and what will be accomplished.

As the Definitive guide states, the Scrum Master is a leader who serves and implies the leadership skills the Scrum Master must have or must acquire. In the case of the Ph.D. student in this study, he has former experience in managing and leading people through his twenty-year career in the military. This fact helped him realize the value of the Scrum framework and allowed easy implementation using only the Definitive Guide. After a two-day Scrum Master training by a registered Scrum practitioner from scruminc.com, we discovered that possessing leadership skills is good but not necessary in applying Scrum in higher education. In general, we firmly believe that no prior knowledge, apart from careful study of the Definitive Guide, is needed to use the Scrum framework successfully.

2.2.2. The Role of Developers

Three teams were created for the first phase, with three students in the first two teams and four students in the third team. The division was determined by the Scrum Master—it was determined that the Scrum Master's (Ph.D. student) division would incorporate different levels of knowledge/experiences in each group since the participants were at another point in their degree program, as Figure 4 depicts. In the second phase, two teams comprised the remaining nine students, with four and five students in each group. An effort was made to mix the members of each group of the first phase with members from other teams to investigate the team interactions and effects on collaborative learning.

2.2.3. The Final Product

The final product of the first phase was a 45-minute presentation explaining the function of an electronic/electric device that relies on the concepts and theory of the backlog items. Each of the three teams had a different device to describe. The Definition of Done by the Product Owner (the professor) was to explain the function of the device using terms and meanings of the related physics course and not with popular science terms. All three teams managed to deliver successfully the above final product, so we had a total of three 45-minute presentations (products) at the end of the first phase.

For the second phase, the final product was a 45-minute presentation of the same electric device, again with the same obligation of explaining the function of the device by

relating its features with the physics principles they have learned. Both teams succeeded in the task, so we had a total of two 45-minute presentations (products) at the end of the second phase.

### 2.2.4. The Product Backlog

The product backlog is the tool for setting the learning outcomes of the project, and the self-organization and self-management of the Scrum Team. It is the drive for the participants to engage actively, not only to the accomplishment of the learning objectives but also for the cultivation of soft skills, along with the project evolution. The framework dictates that the Scrum team along with the stakeholders should create the product backlog in the very first Sprint Planning event. In our case, the backlog was constructed in collaboration with the participant students. However, the main learning objectives of the course (Physics II) mainly dictated the items of the backlog (see Figure 2); the devices to be explained using Physics II concepts from Electronics were proposed, discussed, and decided upon with the students' teams before the announcement of the final backlog list. The long-term reason was to ensure that all the participants were given the correct learning objectives for this project.

PRODUCT BACKLOG

1. Electric Charge
2. Ways for charging bodies
3. Interactions of electric charges
4. Electric charge and the structure of Matter
5. Ions and how they are created.
6. Conservation of electric charge
7. Kirchhoff's Current Law
8. Conductors, Insulators, and Induced Charges
9. Coulomb's Law
10. Scalar and vector quantities.
11. Vector Addition and Subtraction
12. Superposition Theorem
13. Electric Field and Electric Forces
14. Electric Field of a Point Charge
15. Electric Field Lines
16. Charge and Electric Flux
17. Flux of a uniform electric Field
18. Gauss's Law
19. Applications of Gauss's Law-Faraday Cage Function
20. Charges in Conductors-Function of the Lightning Rod
21. The function of Grounding
22. Electric Current
23. Mobility and Conductivity
24. Biot & Savart's Law
25. Gauss's Law in Magnetostatics
26. Ampere's Law
27. AC Current
28. Ampere-Maxwell's Law
29. Faraday's Law
30. Maxwell's Equations
31. Introduction to Waves

**Figure 2.** The Product Backlog.

### 2.2.5. The Sprint Backlog

Each team decided on which items from the Product Backlog they would focus for the next Sprint, and this was the Sprint Backlog. The goal for each group (which is called the Sprint Goal) is to learn, understand, and be able to teach in a simplified manner the items of the Sprint Backlog. The way of achieving this goal was left to the teams along with the Product Owner's advice to also attend the online lectures of the course, as a valuable source of information about their project.

### 2.2.6. The Increments

For the teams to prove that they have gained the necessary knowledge during each Sprint, they were asked to provide a 30-minute presentation in each Sprint Review meeting explaining all the items in the Sprint Backlog in a teaching manner. The criteria for a successful presentation were: (a) all team members must speak during the presentation, (b) every member must be able to answer at least one question about the subjects, and (c) the duration of the presentation must not deviate more than 2 minutes from the assigned time. If the above criteria were met by the team, then the team had successfully produced an Increment.

The purpose for those criteria was to evolve each team member's soft skills in communication, collaboration, critical thinking, and deadline-keeping. For succeeding in the latter, a burndown chart was required in each daily Scrum meeting, to be filled according to the Sprint's work completion. An example of a burndown chart is shown in Figure 3; it documents the estimated effort by the team, for each Sprint, and the progression towards the completion of the Sprint Increment.

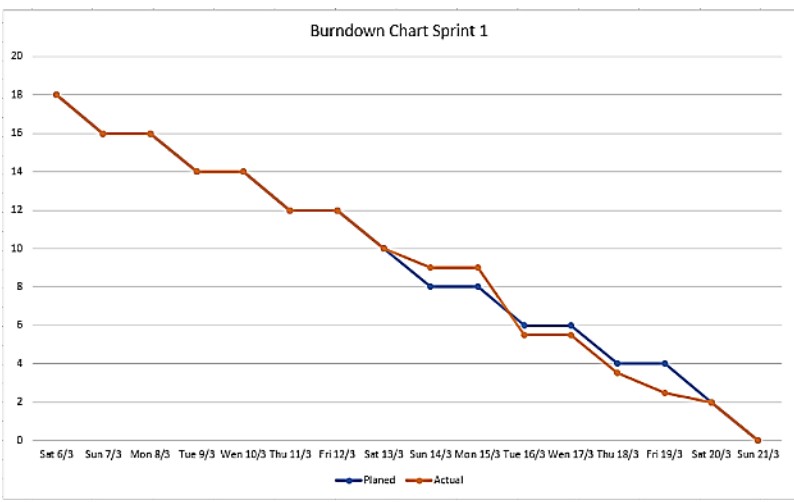

**Figure 3.** Example of a Burndown chart.

### 2.2.7. The Event of Sprint Planning Meeting

This important event for the Scrum framework was organized through a teleconference session on pre-planned dates at the beginning of each new Sprint. For educational purposes and time efficiency, a single meeting was held for all the teams so each participant student could gain valuable experience on how other people think, talk, and behave in an online forum, where important decisions should be made collaboratively and with consensus.

In the first Sprint Planning meeting, all the students decided that the most suitable platform for this online experience was the popular Discord [23]. It is designed for virtual interaction among people with common interests in video games, and it is available for personal computers and intelligent handheld devices, such as smartphones and tablets. It is free of charge, and the overall setup of the application in text and voice channels made it ideal for the project. With the help of the most experienced Discord students, all teams created their own virtual working space, the Scrum server, with virtual rooms inside it that suited the needs of each group. The application offers the possibility of voice and video interaction among members with the click of a mouse button as well as the ability to upload/download any type of digital file up to a specific size, depending on the file type. It also offers the share screen function that proved valuable when one student presented findings on a Product Backlog item to the other team members. Two distinct processes took place in this first Sprint Planning and each subsequent planning. The first was the construction of the Sprint Backlog, in which each team decided which Product Backlog items would be included in this Sprint work. The second and most challenging process was determining how many hours were required to complete each item. The sum of those hours, called the Sprint effort, was displayed on the vertical axis of the Burndown chart, where the calendar days of the Sprint were on the horizontal axis.

### 2.2.8. The Event of Sprint

The first phase was divided into three Sprints of fourteen days per Table 1. The second phase had only one Sprint which lasted three weeks, from the 10 May 2021 until the 28 May 2021. The reason behind scheduling only one Sprint was to discover how they perform in more extended deadlines without the feedback of multiple Sprint Review meetings before the final delivery of the product.

**Table 1.** Start and end dates for the three scheduled Sprints of the first phase.

| Sprint | Start Date | End Date |
|---|---|---|
| Sprint 1 | 8 March 2021 | 21 March 2021 |
| Sprint 2 | 26 March 2021 | 9 April 2021 |
| Sprint 3 | 12 April 2021 | 24 April 2021 |

At the end of each Sprint (except the last one), each team provided the so-called Increment. The Increments produced in the first phase of the project were six PowerPoint presentations that were presented in the first and second Sprint Reviews. Regarding the last Sprint of the first phase, students intergraded the knowledge of all the three Sprints and presented the Final Product.

The second phase had no Increments because there was only one Sprint, with the Final Product delivered at the end of it. However, the Agile character of Scrum remained through Daily Scrum sessions.

The hours assigned by estimation on each Sprint were different for each team as per Table 2. During the Sprint Review at the end of each Sprint, the teams were asked how they estimated the number of hours for the Sprint. All the teams' answers were deliberated, without following any specific methodology, leading Team 1 to estimate more hours than needed and Team 2 to do the exact opposite. Only Team 3 managed to estimate quite accurately the amount of effort required for each Sprint.

**Table 2.** Sprint Effort in Hours for each team on both Scrum phases.

|  |  | Team 1 | Team 2 | Team 3 |
|---|---|---|---|---|
|  | Sprint 1 | 30 | 13 | 21 |
| 1st Phase | Sprint 2 | 35 | 16 | 20 |
|  | Sprint 3 | 26 | 18 | 20 |
|  |  | Team "Generator" | Team "Full House" |  |
| 2nd Phase | Sprint 1 | 22 | 20 |  |

### 2.2.9. The Event of Daily Scrum Meeting

As the official Scrum guide dictates [4], at the beginning of each working day, a daily Scrum meeting with the Scrum Master should take place to track completed and remaining work, interactions, and problems the team may encounter. A parallel objective of Daily Scrum is to guide the students into PowerPoint presentation-making, following proper methods of presenting their work remotely.

Due to the self-managed feature of the Scrum teams, each team decided to work in different time slots during the week to accommodate every team member's need for the other courses in which they were enrolled that semester. The daily Scrums took place at the beginning of each team's working day; the duration depended on the team's needs but never exceeded half an hour. For both phases, each group was encouraged to provide, by email to the Scrum Master, a weekly schedule for the next week, which stated the days and time windows where the project work should take place. This was not a Scrum artifact but solely an aid for coordination between the Scrum Master and the developer teams.

Daily Scrum was conducted each working day on the Discord platform to discuss the three questions of the Daily Scrum. Each team had its own pace of work, which decreased through the first phase of Scrum, as Table 3 shows. At the start of the Daily Scrum, the team presented the updated Burndown chart like the example in Figure 3, where a small discussion was made on the progress of the team's work.

**Table 3.** First phase's working days per Sprint, per team.

| Sprint | Team 1 | Team 2 | Team 3 |
|--------|--------|--------|--------|
| 1st | 6 | 5 | 8 |
| 2nd | 5 | 5 | 6 |
| 3rd | 5 | 3 | 5 |

The preferred method of presenting the chart was the MS Excel spreadsheet, which offered the flexibility of correcting and updating the chart remotely using cloud services. Afterwards, and according to the guide [4], the Scrum Master asked all team members individually, in the presence of the whole team, what they had done the previous working day, what they would do this day, and whether any problems are impeding their work.

The duration of Daily Scrums averaged 30 minutes due to the coaching conducted by the Scrum Master, after finishing the typical Scrum obligation of the three questions to each student. This coaching was deemed necessary to help the students enhance their soft skills of digital literacy, communication, decision-making, critical thinking, research methodology, and time keeping. The way of coaching was through pointing to the right source in the literature for them to investigate and not by directly answering the questions that emerged. As for the parallel objective of teaching the students how to make PowerPoint presentations and how to present them, the guidance was provided incrementally. In each daily Scrum meeting, the Scrum Master offered advice and pointers to suitable sources for research and practice.

The Scrum Master did not supervise each team's work (according to the classical meaning of the term) because the primary objective to which we adhered was the cultivation of self-learning, self-managing, collaborative learning, and engagement, actions that lead directly to the active learning concept [9]. He only helped them in aligning with the proper research direction, allowing for some minor deviations for educational purposes. In addition to that, the Scrum Master provided incentives and posed theoretical questions to the teams to improve their critical thinking and analysis from one Daily Scrum meeting to the next.

### 2.2.10. The Event of Sprint Review Meeting

The event of the Sprint Review was held in Zoom online tool, with all the teams virtually present during each team's 30-minute Increment presentation. The aim of each presentation was for each team to demonstrate their level of understanding of the Backlog Items they had chosen in a teaching–lecture manner.

The Developers were encouraged to choose their own way of presenting their work but with a set of criteria provided by the Product Owner. Through these assessment criteria, the students were able to prepare their presentations themselves and without the intervention of the Scrum Master or the Product Owner. Each online presentation was followed by a 30-minute Question and Answer period, where the professor asked each team member key questions about the content of their presentation to assess their level of understanding of each physics law or concept that was presented and is linked with the operation the device was targeted. Members from other teams were permitted to ask questions of the presenting group, exchange ideas, and provoke a discussion about the presentation.

### 2.2.11. The Event of Sprint Retrospect Meeting

This important event for the team's evolution was held the day following the Sprint Review, separately for each team, and the focus was on how the team members interacted during the Sprint. In addition, there was a discussion on how the group worked regarding Product Backlog items research and handling. With the encouragement of the Scrum Master, the students decided, by consensus, whether the methods and tools used in the previous Sprint were suitable for the next one.

### 3. Results

As stated earlier in this paper, this effort was additional and instrumental to the conventional attempt of the class taught online. The end of the project coincided with the end of the semester when the Scrum students undertook the final exams along with the rest of the enrolled students. The grading system at the Hellenic Mediterranean University, where this study was conducted, qualifies a score greater than five out of ten as a passing score. The comprehensive exam results of the entire class are shown in Table 4. For comparison reasons, the enrolled students were divided into two categories, those who participated in the Scrum project and those who did not.

**Table 4.** Exams results of the Physics II course.

|  | No SCRUM | SCRUM |
|---|---|---|
| No. of final exam participants | 92 | 8 |
| No. of participants who passed | 31 | 5 |
| Success rate | 34% | 63% |
| Average final exam grade | 3.8 | 6.7 |

After the end of the semester, the participants were asked for feedback on the Scrum experience through an online questionnaire with 23 questions. The most critical responses received are shown in Figures 4–7.

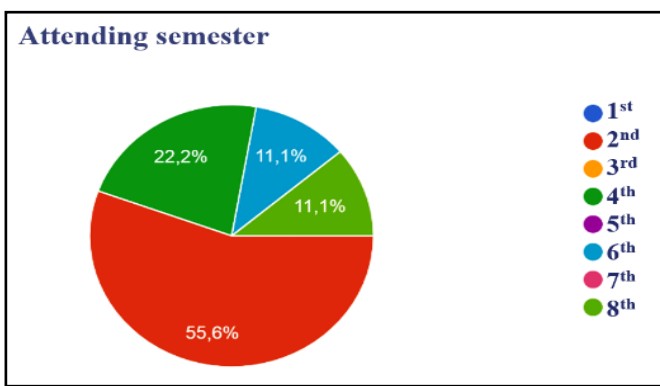

**Figure 4.** Attending semester distribution among Scrum participants.

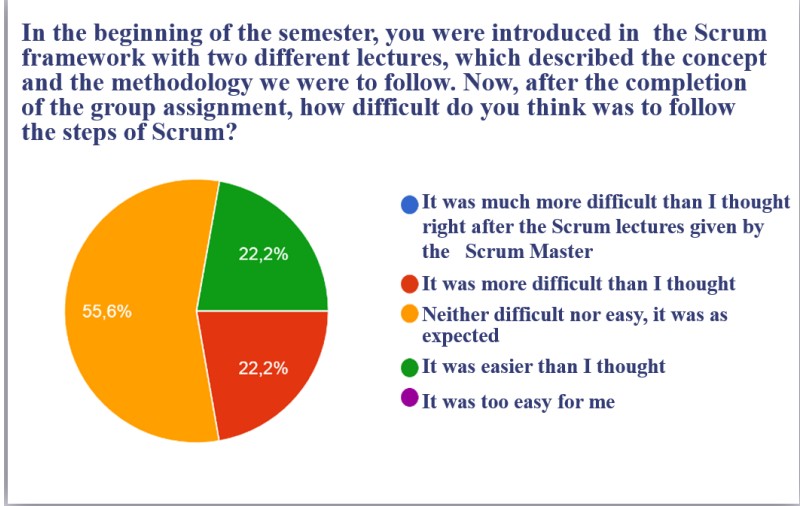

**Figure 5.** Perceived difficulty of Scrum after the completion of the project.

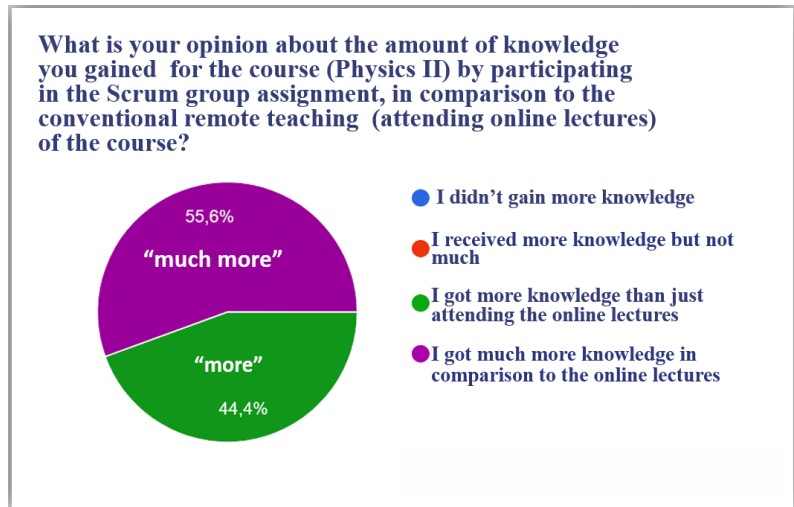

**Figure 6.** Perceived gain of knowledge with Scrum compared with the conventional online lectures.

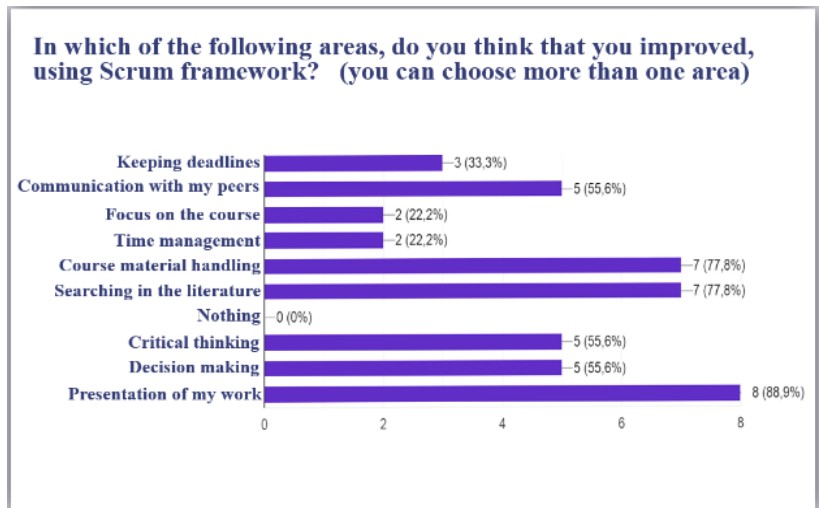

**Figure 7.** Perceived soft skills enhancements.

As is shown in Figure 4, most of the participants were from the second semester, where the Physics II course is taught for 13 weeks, while the remainder of the students were further along in their degree and had been enrolled in the course before. In Greek universities, the students are allowed to advance through their degree, even though they have failed to pass courses from previous semesters.

The Scrum facilitators (Scrum Master and Product Owner) tried to discover how difficult it was for the students to apply the framework by asking the question in Figure 5.

Figure 5 responses were expected since this was the first time the students have been exposed to this type of disruptive (for them) learning methodology. This methodology entailed, for the students, a high level of commitment, critical thinking, collaboration, and planning compared with the traditional teaching methods to which they had been exposed until now. We strongly believe that since Scrum will be applied by more teachers in the future, the students will feel more comfortable with this process.

The most crucial question was whether the Scrum helped the participants to better learn the course material compared with just watching the online lectures. The students' responses are depicted in Figure 5.

In the pursuit for the secondary goal in this study, it was important for the facilitators (Product Owner and Scrum Master) to know what the students thought about their enhancement in soft skills. The corresponding replies are shown in Figure 7.

During the semester, the professor evidenced the progression of their acquired knowledge through their responses to the oral questions he posed to each student individually in every Sprint Review. The questions were according to the course's learning outcomes, in an increasing difficulty manner throughout the progression of the project. This fact was ultimately proved true from the exam results of the students who participated in the Scrum project, as shown in Table 4. In addition to that, the grades the Scrum students achieved were higher in percentage than the grades of the rest of the class.

## 4. Discussion

We believe, to the best of our knowledge, that this is the first time Scrum methodology is transferred and applied in the context of a STEM course. This showed the versatile applicability of the framework in theoretical and experimental science courses, beyond ICT solely. The Scrum framework consists of a very effective training method of soft skills development. Moreover, the much higher success rate of the students participating in the collaborative learning approach through Scrum (see Table 4) provides sound evidence of the framework's success.

The students' engagement in the Scrum project was apparent from the first Sprint Planning, where they cooperated to decide which platform to use for their meetings and work. This engagement remained systematic throughout the project since there were only two instances where a student of each team was absent for the Daily Scrums of the first phase. The three Scrum questions posed by the Scrum Master in each Daily Scrum to each team member provided the Scrum Master with a way to document the progression of each member on embracing the values of Scrum. Although in the beginning, it was difficult for some students to express themselves openly, with the frequent iteration of the Daily Scrums, the value of openness was increasingly embraced by all the students. This openness led them to receive and give respect, which, in turn, gave them the courage to admit shortcomings and personal deficiencies that affected the team's work. However, the most evident evolution in the students' behavior was their commitment to the task during the project. This commitment and focus were observable through the frequent virtual visits of the Scrum Master on the Discord Platform, where he could silently watch the teams working in their respective space.

Another difficult task was predicting the work needed for the Sprint Backlog items. Two teams were unsuccessful in precisely estimating their effort at the beginning of each Sprint, but this was not a problem since the dynamic nature of Scrum permits corrections and adaptations to every action made towards the Sprint Goal. Both the Product Owner and the Scrum Master witnessed the students' performances in the presentation of their work on each Sprint Review, which varied from adequate in the first meetings to very good at the end of the project. The evolution of their presentation and time management skills were due to the repetitive process of Sprint Reviews and the feedback from the Product Owner, who was also the professor delivering the course.

Although the teams did not produce the entire Product Backlog, the artifact of Sprint Backlog was correctly created by all the teams since they managed to deliver the backlog items at the end of each Sprint. The last artifact of the Sprint Increments, which was the half-hour presentation, was evaluated to be of ascending quality by the Product Owner and the Scrum Master.

In the second phase of the Scrum, where there was only one Sprint with 18 calendar days, the performance of the teams was not affected, even though they had only one Sprint Review meeting at the end.

## 5. Conclusions

This work was a qualitative study (the limited number of participants does not allow us to define it as a quantitative-based work) of the application and impact of Scrum in a Physics course rigorously following the Scrum framework [4]. The authors strongly believe that this framework can be applied without extra training for the Scrum Master, as long as

the entire Scrum Team adheres to and embraces the pillars and values of Scrum as a first step and then follows the events of Scrum.

The five values of Scrum enabled the participants to evolve on a personal level as collaborators towards the common task. The self-managing feature of the framework cultivates, in the students, the soft skills of communication, cooperation, creativity, and problem-solving, which are essential for their employability.

The pillars of Transparency, Inspection, and Adaptation lead to the evolution of work ethic and attention to detail. The iterative nature of Scrum trains students on timekeeping in an easy and understandable way.

New skills can be acquired from the Developer's team if the Scrum Master uses the Daily Scrum as an opportunity for inspiration and motivation. However, the Scrum Master may exceed the time limit of 15 minutes, as described in the guide.

In theoretical courses such as Physics II, although the limitations of the learning outcomes of the course are set, the precise application of the Scrum guide can be implemented. In our case and during the formulation of the product backlog, we engaged students in the proposition that electronic devices can be explained using the Physics II concepts. Our experience also showed that if a trainer in a theoretical course elects to follow the method of giving a readymade Product Backlog, this could also work well for the learning outcomes of the curriculum. This is not considered to be a deviation from the Scrum guide, provided that the readymade backlog is constructed by the Product Owner who is solely responsible for the final product and the fulfillment of the stakeholders' expectations.

Concerning the duration of the Sprints, the authors suggest two weeks duration for the teams to receive more frequent critical feedback from the Product Owner.

Among the authors' future actions is the application of the Scrum framework for all the course participants. This way, it can be determined how effective the framework is in teaching students to be committed to the course, design their study process, teach each other independent of their learning motivation level, and work in groups. Possible caveats to this endeavor may be the team size and the number of Scrum Masters needed for the entire class. With only a few Scrum Masters, the team size could reach the maximum length of ten students, which will pose a drawback in team communication. Another possible obstacle Scrum facilitators may face in such a case is the lack of motivation from students since the results presented here were based on volunteers only. Added to that, the additional workload of projects with Scrum throughout the semester may prevent the students from committing to the project. At the institutional level, the highest challenge that a Scrum facilitator may encounter is the unwillingness of the higher education institution to embrace the pillars and values of Scrum towards more engaging and effective online or face-to-face teaching using group projects. However, the application of Scrum from a small group of students to the general classroom and the expected impact the framework has on students' course success and understanding will convince the rest of the trainers to apply the proposed pedagogy in other curriculum modules.

**Author Contributions:** Conceptualization, E.L. and K.P.; methodology, E.L.; validation, K.P. and E.L.; formal analysis, K.P. and E.L.; investigation, E.L.; writing—original draft preparation, E.L.; writing—review and editing, E.L. and K.P.; visualization, E.L.; supervision, K.P.; project administration, E.L.; funding acquisition, K.P. All authors have read and agreed to the published version of the manuscript.

**Funding:** This research received no external funding.

**Informed Consent Statement:** Informed consent was obtained from all subjects involved in the study.

**Conflicts of Interest:** The authors declare no conflict of interest.

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
