# Peer review of "Applying Scrum in an Online Physics II Undergraduate Course: Effect on Student Progression and Soft Skills Development"

_education, doi:10.3390/educsci13020126_

Round 1
Reviewer 1 Report
Analyzing the paper, I encountered several issues:
1. The authors speak about “feedback collected from the teachers and the students involved”, but the study is carried out by 2 teachers that are both observing and included in the experiment. Please, clarify how the feedback from teachers was achieved.
2. The author state “The primary goal was to investigate the feasibility of Scrum under remote teaching conditions as a facilitator of collaborative group learning”. But, in the results section, the table 4 and figure 4 present data that are not supporting this, moreover the results are inconclusive. The Table 4 presents the average grades on the two groups (with and without Scrum) after specifying that the passing of exams is for above 5 from 10 points. For all that, the average is below 5 for all cases. The figure 4 the percentages are equally distributed among the responses to “Is was more difficult...” and “It was easier…”. The other only response was “Neither difficult nor easy”. Please, point out the relevance!
3. At page 6: The Product Backlog= The Sprint Backlog, word by word.
4. Please, explain clearly how the Scrum experiment that was an extra activity on the top of the academic experience fits in the total number of hours for individual study for the students and does not affect the amount of time allocated to other courses.
5. In the figure 2, Burndown chart for sprint 1, but it does not match any previous data, sprint 1 starts both on 8th of March (table 1) and 6th of March (figure 2). Also, the estimated hours at the beginning do not match.
6. There are not clearly specified the PowerPoint presentations. In phase 1 there are 6 PowerPoint presentations for first 2 sprints and nothing for the third and in Phase 2 there are 5 PowerPoint presentations for 2 teams in one sprint.
7. The 65 burndown charts are not correlated in the paper to the students’ daily activities. Why 65?
8. How relevant is the research if the authors say that the whole Product Backlog was not produced?
9. What does it mean “The authors are motivated by Scrum …” (line 83)?
Reviewer 2 Report
Review Report
Article title: Applying Scrum in an Online Physics II Undergraduate course: Effect on Student Progression and Soft Skills Development
The manuscript describes how Scrum was used remotely on an online Physics II undergraduate course during the pandemic lockdown period of Electronics Engineering Department at the Hellenic Mediterranean University of Greece. The primary goal was to investigate the feasibility of using the Scrum approach under remote teaching conditions to facilitate a collaborative group learning. A secondary objective was to observe the students’ interactive engagement in distance learning, along with the possible advancement of soft skills.
Specific comments:
1. The research questions and research hypotheses being tested need to be clearly mentioned at the end of the “Introduction” section.
2. The manuscript presents the individual phases and processes of Scrum in a very general way. A concrete application of Scrum to a specific topic in online physics education should be presented.
3. The section “Discussion” needs to be better unfolded, highlighting what is new in the research and how the outcomes of the study can be used in practice. The authors should discuss the results and how they can be interpreted in terms of research hypotheses. Have the results been compared with literature? Have you found any similarities or discrepancies with previously published data? What are the benefits of the study for the future?
Round 2
Reviewer 1 Report
Accept in present form
Reviewer 2 Report
Dear Authors,
I appreciate the great efforts that you have made in response to my questions and comments. The revision clarifies all the points I raised. You have significantly improved the clarity of your writing and have addressed most of my concerns.
Kind regards,
The Reviewer